# Comprehensive Lifestyle Modification Influences Medium-Term and Artificially Induced Stress in Ulcerative Colitis—A Sub-Study within a Randomized Controlled Trial Using the Trier Social Stress Test

**DOI:** 10.3390/jcm10215070

**Published:** 2021-10-29

**Authors:** Anna K. Koch, Margarita Schöls, Heidemarie Haller, Dennis Anheyer, Zehra Cinar, Ronja Eilert, Kerstin Kofink, Harald Engler, Sigrid Elsenbruch, Holger Cramer, Gustav Dobos, Jost Langhorst

**Affiliations:** 1Department of Internal and Integrative Medicine, Evang. Kliniken Essen-Mitte, Faculty of Medicine, University of Duisburg-Essen, 45276 Essen, Germany; a.koch@kem-med.com (A.K.K.); m.schoels@kem-med.com (M.S.); H.haller@kem-med.com (H.H.); d.anheyer@kem-med.com (D.A.); zehra-cinar@gmx.de (Z.C.); r.m.eilert@web.de (R.E.); kerstinkofink@gmx.de (K.K.); h.cramer@kem-med.com (H.C.); g.dobos@kem-med.com (G.D.); 2Department of Internal and Integrative Medicine, Klinikum Bamberg, Sozialstiftung Bamberg, 96049 Bamberg, Germany; 3National Centre for Naturopathic Medicine, Southern Cross University, Lismore, NSW 2480, Australia; 4Institute of Medical Psychology and Behavioral Immunobiology, University Hospital Essen, University of Duisburg-Essen, 45122 Essen, Germany; Harald.Engler@uk-essen.de; 5Department of Medical Psychology and Medical Sociology, Faculty of Medicine, Ruhr University, 44801 Bochum, Germany; Sigrid.Elsenbruch@ruhr-uni-bochum.de; 6Department for Integrative Medicine and Translational Gastroenterology, Medical Faculty, University of Duisburg-Essen, Sozialstiftung Bamberg, 96049 Bamberg, Germany

**Keywords:** stress, ulcerative colitis, comprehensive lifestyle modification, Trier Social Stress Test, clinical research

## Abstract

Objective: The present study presents long-term results of stress-related outcomes of a prospective RCT that evaluated effects of a ten-week comprehensive lifestyle-modification program (LSM) in patients with inactive ulcerative colitis (UC). In addition, exploratory results of a sub-study applying a laboratory stress protocol (Trier Social Stress Test; TSST) conducted within the RCT are reported. Methods: Ninety-seven patients with inactive UC were randomized to LSM (*n* = 47; 50.28 ± 11.90 years; 72.3% female) or self-care (*n* = 50; 45.54 ± 12.49 years; 70% female). Patients’ perceived stress, anxiety, flourishing and depression were assessed at week 0, 12, 24 and 48. After the respective intervention, 16 female patients (LSM: *n* = 8; 44.6 ± 14.3 years; Self-care: *n* = 8; 49.25 ± 4.30 years) additionally underwent the TSST. State anxiety, blood pressure, pulse, complete blood counts, adrenocorticotropic hormone (ACTH), cortisol, adrenalin and noradrenalin were measured at *baseline* (−15 min), *stress* (+10 min), *recovery1* (+20 min) and *recovery2* (+55 min). Statistical significance was set at *p* < 0.05; for the exploratory sub-study using the TSST, *p*-values < 0.10 were considered significant. Results: Patients’ perceived stress declined significantly after the LSM (*p* < 0.001) compared with control. This lasted until week 24 (*p* = 0.023) but did not persist until week 48 (*p* = 0.060). After 48 weeks, patients’ flourishing was significantly increased compared with control (*p* = 0.006). In response to the TSST, significant group differences were evident for pulse (*p* = 0.015), adrenaline (*p* = 0.037) and anxiety (*p* = 0.066). After 55 min, group differences were found for ACTH (*p* = 0.067) and systolic blood pressure (*p* = 0.050). Conclusions: LSM has a medium-term positive effect on perceived stress. First indications show that it is promising to investigate these effects further under laboratory conditions. It is also desirable to find out how the effects of LSM can be maintained in the long term.

## 1. Introduction

The broad role of psychosocial stress in inflammatory bowel diseases (IBD) has been shown both in animal experiments and in human studies [1,2,3,4,5,6]. Although there is consensus that stressful life events, mental stress and psychological disorders are not causative for the development of ulcerative colitis, stress demonstrably has a negative impact on the severity and course of the disease. Furthermore, a large proportion of patients report that stress has a negative impact on the progression of their disease and that stress has already triggered a flare [7,8]. This can also result in a vicious circle: disease management problems cause stress, and stress then triggers IBD episodes [9,10]. Patients who have a high level of perceived stress also have a reduced quality of life [11]. An adaptive approach to improved coping with stress and stressful life events thus presents a challenge for the patients, with possible implications for disease course and well-being.

There are studies on acute stress [12,13,14], but not yet on interventions that address this very issue. For example, a study by Agostini et al. (2017) provided insight into the relationship between stress and brain functional changes in patients with Crohn’s disease using functional magnetic resonance imaging [12]. Seventeen Crohn’s disease patients and seventeen healthy controls underwent an fMRI scan while performing a stressful task consisting of a Stroop color and word interference task. Compared with controls, the stress task triggered greater blood oxygen level-dependent signals in the midcircular cortex in CD patients. Further studies have shown that activation of mast cells in the mucosa and damage to the epithelium as induced by an artificial stressor are more pronounced in IBD patients than in healthy controls, which might trigger relapses [14]. However, research calls for studies on interventions that might help to answer the question whether stress contributes to altered symptom perception via learning and memory processes [15].

Mind–body techniques such as yoga, mindfulness and autogenic training have been shown to have positive effects on stress regulation in healthy persons and persons with gastrointestinal diseases [16,17,18,19]. There are various psychological and biological mechanisms via the brain-gut axis that could explain such therapeutic effects that, however, have not been tested yet, for example, through altered gene expressions that counteract cellular damage related to chronic stress. Even short-term mind–body practice induced those changes in gene expression [16,17]. Further modes of action might include activation of the sympathetic nervous system and mast cells, vagal inhibition, prefrontal-amygdala complex and the immune system, downregulation of the hypothalamic corticotropin-releasing factor (CRF) system, the peripheral CRF system in inflammation, early life events and effects of depression [20]. Comprehensive lifestyle-modification programs include different aspects of mind body-medicine and aim to improve patients’ health and quality of life and to teach adaptive coping with stressful events [21,22,23]. The comprehensive lifestyle modification used in this study is tailored to the needs of patients with ulcerative colitis and includes ten weekly sessions of 6 hours each, resulting in 60 h total. During the sessions, patients learn about mind–body medicine with aspects of mindfulness, progressive muscle relaxation, body scan, exercise such as walking and yoga, self-care strategies, stress management, dietary counselling with focus on the Mediterranean diet, naturopathy self-help strategies and herbal medicine.

After the ten intervention weeks, a sub-sample of the patients participated in the *Trier Social Stress Test* (TSST), which is the gold standard for inducing acute psychosocial stress under laboratory conditions [24]. Hence, in the present study the effects of a comprehensive lifestyle-modification program on different stress parameters in patients with ulcerative colitis were investigated exploratively within a sub-study of a randomized controlled trial (RCT). Accordingly, this study pursues two goals: first, to analyze long-term effects of the intervention, especially regarding stress, using standardized questionnaires in the overall sample and second, to compare the effects on TSST in a sub-sample. To our knowledge, no randomized controlled trial has yet evaluated the effects of the comprehensive lifestyle-modification program on acute stress under laboratory experimental conditions. 

## 2. Materials and Methods

### 2.1. Study Design

The present study presents long-term results of stress-related outcomes of a prospective RCT [25]. In addition, exploratory results of a sub-study applying a laboratory stress protocol conducted within the RCT are reported. The study was conducted at the Evang. Kliniken Essen-Mitte, Germany, between 2016 and 2019, approved by the Ethics Committee of the University of Duisburg-Essen (approval number 15-6554-BO), registered on clinicaltrials.gov (clinicaltrials.gov ID: NCT02721823, https://clinicaltrials.gov/ct2/show/NCT02721823 (accessed on 1 June 2021)) and performed according to the declaration of Helsinki, applying up-to-date good clinical practice standards. The previously published results report the effects of the comprehensive lifestyle-modification program on disease activity and quality of life [19]. The present manuscript aims to answer the question whether patients who had undergone the intervention program differed in their responses to stress from the patients who were in the control group. Various subjective stress questionnaires were collected before and after the intervention. Additionally, female patients from both groups voluntarily underwent the TSST shortly after the intervention group completed the program. Regarding the TSST, all patients were informed about the experiment and could choose whether they wanted to participate. Due to the different hormonal responses to stress between the sexes, only female patients were included in the TSST subsample. All patients gave written informed consent.

### 2.2. Randomization

Patients were randomized using computer generated stratified block randomization (Strata: sex and azathioprine) to either the lifestyle-modification group or the control group using the Random Allocation Software. For further information see Langhorst, Schöls [25].

### 2.3. Participants

Within the RCT, all patients underwent a medical visit on site at the beginning of the study. Patients with diagnosed but currently inactive ulcerative colitis (Clinical Activity Index according to Rachmilewitz (CAI) ≤ 4), between the age of 18 and 75 years, and impaired quality of life (Inflammatory Bowel Disease Questionnaire total score < 170 at baseline) were included. Exclusion criteria were: Infectious or chronically active colitis, glucocorticosteroids taken within the last 3 months, except for stable medication with azathioprine (other medicament treatments according to the medical guideline are allowed, for example by mesalazine or sulfasalazine), severe psychological illness (e.g., depression, addiction, schizophrenia requiring treatment), severe comorbid somatic disease (e.g., diabetes mellitus, oncological disease), pregnancy or participation in a stress-reduction program or other clinical studies on psychological interventions. Exclusion criteria were checked within a medical interview.

Regarding the TSST, due to the different hormonal responses to stress between the sexes [26], only female patients were approached for participating in the TSST. Participation was voluntary.

### 2.4. Intervention and Control

Patients in the comprehensive lifestyle-modification program including mind–body techniques participated in ten 6-hour group sessions within ten weeks. For further information on the intervention, see Langhorst, Schöls, Cinar, Eilert, Kofink, Paul, Zempel, Elsenbruch, Lauche, Ahmed, Haller, Cramer, Dobos and Koch [25].

The control group participated in a one-time workshop with intensive training in naturopathic self-help strategies by an experienced integrative medicine gastroenterologist with various self-care strategies, herbal medicines and home remedies being presented.

### 2.5. Long-Term Measures

Questionnaires were completed before (week 0), post intervention (week 12) and at two follow-ups (week 24 and week 48). *Perceived stress* was measured using the 10-item German version of the Perceived Stress Scale (PSS-10) [27,28]. Higher scores indicated higher perceived stress. Answers could be given on a 5-point Likert scale (0 = *never*, 4 = *very often*). *Anxiety* and *depression* were assessed using the Hospital Anxiety (HADS-A; seven items) and Depression (HADS-D; seven items) Scale, with higher values indicating higher anxiety or depression, and values > 8 indicating potential sub-clinical anxiety or depressive disorders [29,30]. Answers were rated on a 5-point Likert scale (0 = *best rating*, 4 = *worst rating*). Flourishing was measured with the Flourishing scale which contains eight items that could be rated on a 7-point Likert scale (1 = *worst rating*, 7 = *best rating*) [31,32]. Hence, the total score ranges from 8 to 56, with a high value describing a person with many psychological resources and strengths.

### 2.6. Sub-Study: Trier Social Stress Test

#### 2.6.1. Procedure

The TSST is a standardized protocol for the induction of moderate psychosocial stress in a laboratory setting [33]. The patients were invited to the Department of Internal and Integrative Medicine, Evang. Kliniken Essen-Mitte, Germany, between 1 and 5 pm. An indwelling venous cannula was placed, and patients rested for 30 min. At 15 min before the patients were taken to the test room where the TSST took place, questionnaires were filled in, the first blood sample was taken and the blood pressure and pulse were measured (Time point *Baseline*). Afterwards, a neutral experimenter led the patients from the preparation room to the test room where the stress test took place (see Kirschbaum, Pirke and Hellhammer [33] for a detailed description). After the stress test, the second questionnaire, blood sample, blood pressure and pulse were taken directly (Time point *Stress*). Thereafter, the patients were taken back to the preparation room. After 20 (Time point *Recovery 1*) and 55 (Time point *Recovery 2*) minutes, the third and fourth questionnaire, blood sample, blood pressure and pulse were taken. Following recovery 2, patients were debriefed and discharged by a study nurse.

#### 2.6.2. Measures

All measures within the TSST were assessed at four time points: *Baseline* (−15 min), *Stress* (+10 min), *Recovery 1* (+20 min) and *Recovery 2* (+55 min). *State anxiety* was assessed with the state version of the State-Trait Anxiety Inventory (STAI-S). The inventory includes 20 items that can be rated on a scale from *1* = *not at all* to *4* = *very much so* [34]. Blood pressure and pulse were assessed using a blood pressure monitor (Visocor OM50; UEBE Medical GmbH, Wertheim, Germany). Venous blood was collected in tubes containing EDTA (S-Monovette, Sarstedt, Nümbrecht, Germany). Complete blood counts including white blood cell differential were analyzed by the local clinical laboratory. Plasma was separated by centrifugation (2000× *g*, 10 min, 4 °C) and stored at −80 °C until analysis. Plasma levels of adrenocorticotropic hormone (ACTH) and cortisol were measured by enzyme-linked immunosorbent assay (ACTH and Cortisol ELISA, IBL International, Hamburg, Germany). The sensitivity of the assays was 1 pg/mL for ACTH and 5 ng/mL for CORT. Cross-reactivity of the anti-ACTH antibody with other peptide hormones was <0.001%, cross-reactivity of the anti-CORT antibody with other relevant steroids was 7.0% (11-deoxycortisol), 4.2% (cortisone), 1.4% (corticosterone), 0.35% (progesterone), and <0.01% (estrone, estradiol, estriol, testosterone). Plasma concentrations of adrenaline and noradrenaline were measured by ELISA (CatCombi, IBL International Germany) following affinity gel extraction, acylation, and enzymatic derivatization. The sensitivity of the assay was 8 pg/mL for adrenaline and 20 pg/mL for noradrenaline. Cross-reactivity of the antiserum with other catecholamines and their metabolites was <0.4%.

#### 2.6.3. Sample Size Calculation and Statistical Analyses

Within the main study, sample size was determined based on the primary outcome *health-related quality of life* [25]. Here, a sample size of 92 was deemed sufficient. To analyze the long-term effects of the comprehensive lifestyle-intervention, especially regarding stress, this overall sample was analyzed with the *p*-value set at *p* < 0.05. These analyses complement the already published manuscript [25]. Analyses were conducted on an intention-to-treat basis. Missing values were replaced by multiple imputation with 50 generated and averaged additional data sets. To compare the effects on TSST, a sub-sample (*n*= 16), was analyzed with the *p*-value set at *p* < 0.10 but with at a lower confidence level. Baseline group differences were analyzed using Student’s *t*-tests for continuous data and chi-squared tests for categorical data. Univariate analyses of covariance (ANCOVA) with ‘group’ as the between-subject factor and baseline values as covariates were applied to evaluate group differences. Due to the explorative nature of the sub-study and small sample size, no adjusted *p* value for multiple testing was applied. All analyses were performed using the Statistical Package for Social Sciences software (IBM SPSS Statistics for Windows, release 25.0; IBM Corporation, Armonk, NY, USA).

## 3. Results

### 3.1. Participants

A total of 97 patients were randomized to either the comprehensive lifestyle-modification group (*n* = 47) or control (*n* = 50). Of those, 16 female patients participated in the sub-study using the TSST (shown in Table 1; Figure 1).

### 3.2. Perceived Stress, Anxiety, Depression, and Flourishing before and after the Interventions

At week 12, patients in the comprehensive lifestyle-modification program reported a significantly decreased perceived stress compared with control. Anxiety, depression and flourishing were not significantly different between the groups. At week 24, the group differences in perceived stress persisted; at week 48, patients’ flourishing was significantly different between the groups in favor of the intervention group (Table 2).

### 3.3. Responses to the TSST

The TSST induced a rise of stress parameters, expressed as increased state anxiety, cardiovascular activation, increased blood counts and heightened stress hormones, in both groups. Inflammatory and hormonal blood levels in response to the TSST are depicted in Figure 2.

At *recovery 2*, the difference of ACTH was significant between the groups on a ten percent level (F(1, 13) = 3.998, ∆ = 2.526 ng/mL, 95%CI [−0.203–5.255], *p* = 0.067, η^2^_p_ = 0.235). Adrenaline was significantly different between the groups at *stress* (F(1, 13) = 5.399, ∆ = −19.073 pg/mL, 95%CI [−36.806–−1.340], *p* = 0.037, η^2^_p_ = 0.293), but not at *recovery 1* or at *recovery 2*. Patients’ cortisol level, noradrenaline, leukocytes, thrombocytes, lymphocytes, monocytes or neutrophiles did not exhibit significant differences between the groups at none of the time points. Cardiovascular measures increased in both groups in response to the TSST. Significant group differences were evident for patients’ systolic blood pressure at *recovery 2* on a ten percent level (F(1, 13) = 4.661, ∆ = −10.238 mmHg, 95%CI [−20.483–0.007], *p* = 0.050, η^2^_p_ = 0.264) and for pulse at *stress* (F(1, 13) = 7.863, ∆ = 6.993 bpm, 95%CI [1.605–12.380], *p* = 0.015, η^2^_p_ = 0.377). Patients’ diastolic blood pressure was not significantly different between the groups at any of the time points (shown in Figure 3).

State anxiety as measured by the STAI-S is depicted in Figure 4. Both groups report an increase directly after the TSST. Patients’ state anxiety was significantly different between the groups at *stress* on a ten percent level (F(1, 13) = 4.027, ∆ = 9.026, 95%CI [−0.690–18.741], *p* = 0.066, η^2^_p_ = 0.237) but not at *recovery 1* or at *recovery 2*.

## 4. Discussion

This study shows that comprehensive lifestyle modification including mind–body techniques positively influences stress perception in patients with inactive ulcerative colitis. Patients’ perceived stress declined significantly after participating in a ten-week comprehensive lifestyle-modification program compared with control: At week 0, patients’ perceived stress in both groups was markedly higher than that of the German population of the same age [28]. After 12 weeks, the perceived stress of patients in the intervention group decreased close to the normal healthy population. This effect lasted until week 24 but did not persist until week 48. After 48 weeks, patients’ flourishing was significantly increased compared with control. No effects on anxiety and depression were evident. Those findings complement the results of the already published manuscript, where comprehensive lifestyle modification was found to improve the health-related quality of life in patients with ulcerative colitis in remission [25].

Those effects of the lifestyle-modification program on stress perception were partly supported by results of the exploratory sub-study applying the TSST under laboratory experimental conditions. The TSST induced a rise of most measured stress parameters in both patient groups. Directly after the TSST, group differences were evident for pulse and adrenaline as well as for anxiety on a ten percent level. Interestingly, whereas pulse and anxiety were lower in the lifestyle-modification group, adrenaline was higher in the intervention group. After 55 min, significant group differences were evident for ACTH and systolic blood pressure. However, due to the very small sample size, these results are only preliminary indications that it is promising to investigate this issue further. The findings are in line with previous research where neuroendocrine stress response to the acute stressor in healthy subjects who underwent the stress management program before the TSST was evident [35]. For patients with IBD, a stress management program that included relaxation techniques, personal planning skills and communication skills influenced disease activity and stress significantly up to one year after the intervention [36]. Some studies, however, did not support those findings. A study that evaluated immune responses and the in vitro adrenergic and glucocorticoid modulation of cytokine production by peripheral blood cells in women with ulcerative colitis compared with healthy controls as induced by a public speaking stress found no significant differences between groups. However, patients showed reduced baseline IFN-gamma production and lower basal cortisol and prolactin levels [37]. Another study on the effects of lifestyle modification on ulcerative colitis found improvements in health-related quality of life but not regarding stress-related parameters such as cortisol, noradrenaline, adrenaline or the distribution of circulating lymphocyte subsets [21].

Hence, given that psychosocial stress plays a critical role in the course and manifestation of ulcerative colitis, techniques to deal with stressful events can help patients to cope with their disease [1,2,3,4,5]. One way to constructively deal with stress is to learn techniques that enable patients to relax and hence experience a so-called *relaxation response*. The relaxation response is seen as the physiological counterpart of the stress response and acts as a protection against potentially harmful stressors [9]. This physiological mechanism is initiated when a person engages in a repetitive physical or mental activity while ignoring distracting thoughts [9]. Even short-term mind–body practice such as yoga, body scan, and autogenic training can enable a person to experience this relaxation response and the associated downregulation of inflammation associated genes [16]. Within the comprehensive lifestyle-modification program, the patients learned many different techniques for stress management and dealing with stressful situations such as mindfulness, progressive muscle relaxation, body scan. Although we have not explicitly explored this in the present study, it is conceivable that the patients experienced a relaxation response using these techniques. This would also complement the at first glance contra-inductive result that patients in the intervention group had higher adrenaline levels: Patients in the lifestyle-modification group exhibited higher adrenaline compared with control directly after the TSST. This finding might seem paradox at first but is in line with previous studies that found a reduced sympathetic nervous system responsivity associated with the relaxation response [38]. This suggests that in patients who were relaxed, i.e., experienced a relaxation response, more adrenaline is needed to produce normal increases in pulse and blood pressure which is supposedly due to a reduced adrenergic end-organ responsivity. These differences between the groups are particularly interesting because previous research suggested that in patients with inactive ulcerative colitis, the downregulation of proinflammatory cytokine production in peripheral blood immune cells through glucocorticoid, adrenergic and cholinergic mechanisms is as efficient as in healthy individuals [6]. Further studies have shown that the activation of mast cells in the mucosa and the damage to the epithelium by an artificial stressor are more pronounced in IBD patients than in healthy controls, which could lead to relapses [12]. It would therefore be interesting to test in future studies whether this activation of the mast cells can also be shown by the TSST and whether the patients who have received the LSM program might react differently here. With regard to irritable bowel syndrome, it has already been shown that cognitive behavioral treatment promotes extinction learning [39,40]. If this could also be shown for ulcerative colitis, this would be the first indication that extinction learning could also be an efficient treatment approach for ulcerative colitis. Strengths of the present study include the randomized controlled design and the administration of a validated stress protocol. The TSST is the gold standard for inducing acute stress under experimental conditions [24]. Furthermore, the measurement of subjective as well as objective stress associate indicators provides a comprehensive overview of patients’ stress-related processes. A limitation of the study is that only female patients participated in the TSST, which was due to the fact that hormonal responses to stress differ between the sexes [26]. A further limitation is the small sample size of the sub-study, which does not allow for firm conclusions to be made. Future studies examining the influence of sex regarding stress parameters in patients with ulcerative colitis are needed. Further, the patient clientele consisted of a sub sample of the RCT. For the RCT, hard inclusion criteria were estimated, which led to an inclusion only of patients with inactive disease and reduced quality of life. Comorbidities such as psychological illness or other somatic diseases were excluded. Hence, the generalizability of the results is limited to a very specific patient clientele. Additionally, because the present study is an explorative sub-study of a RCT, no sample size calculations and no *p*-value adjustments were applied. Due to these limitations, the TSST results have to be interpreted with caution and can only be understood as initial indications that this field of research should be pursued further.

In sum, our findings indicate that comprehensive lifestyle modification may have an impact on the perception and management of stress in patients with inactive ulcerative colitis. Further research is needed to underpin those assumptions and to add to the generalizability of findings, especially in relation to the TSST.

## Figures and Tables

**Figure 1 jcm-10-05070-f001:**
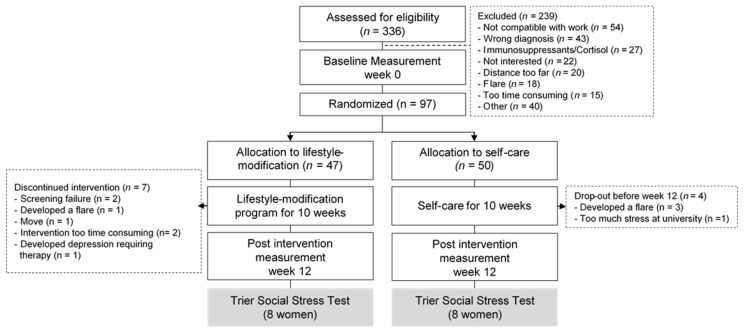
Study flow-chart.

**Figure 2 jcm-10-05070-f002:**
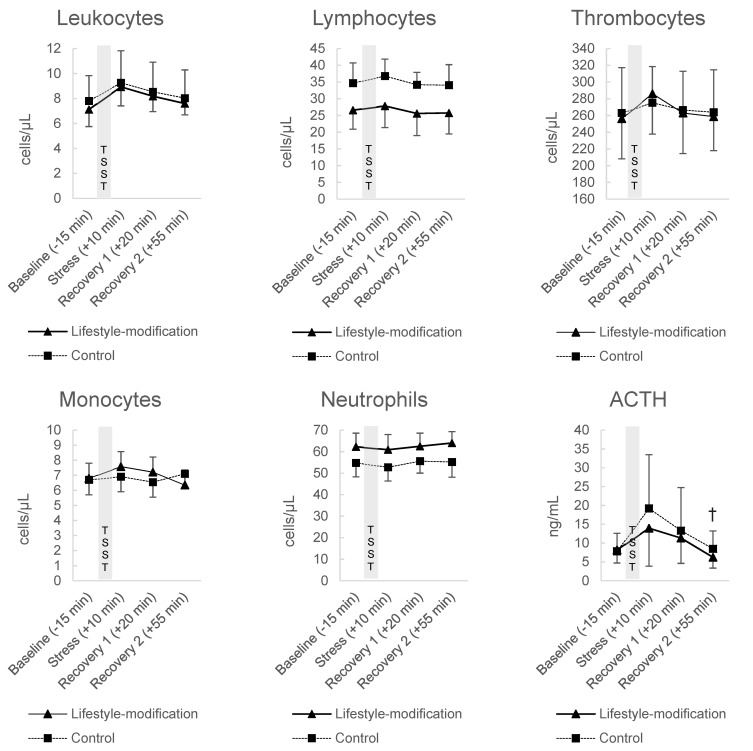
Results of the exploratory sub-study: Blood cell counts and hormonal blood levels in response to the Trier Social Stress Test in the comprehensive lifestyle-modification group (*n* = 8) and controls (*n* = 8). Values expressed as mean ± standard deviation. * *p* < 0.05; † *p* < 0.10.

**Figure 3 jcm-10-05070-f003:**
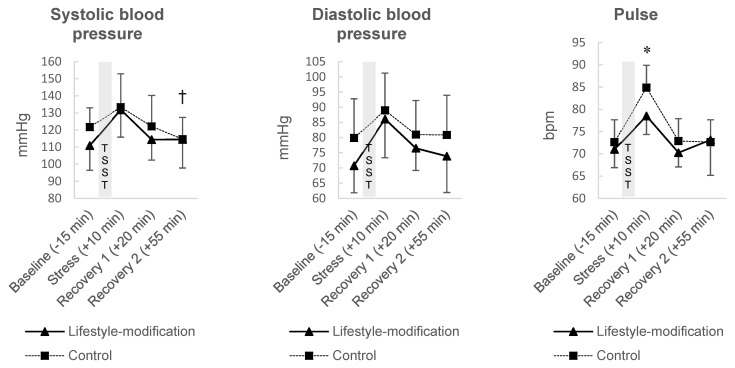
Results of the exploratory sub-study: Cardiovascular responses to the Trier Social Stress Test in the comprehensive lifestyle-modification group (*n* = 8) and controls (*n* = 8). Values expressed as mean ± standard deviation. mmHg = millimeter of mercury; bpm = beats per minute. * *p* < 0.05; † *p* < 0.10.

**Figure 4 jcm-10-05070-f004:**
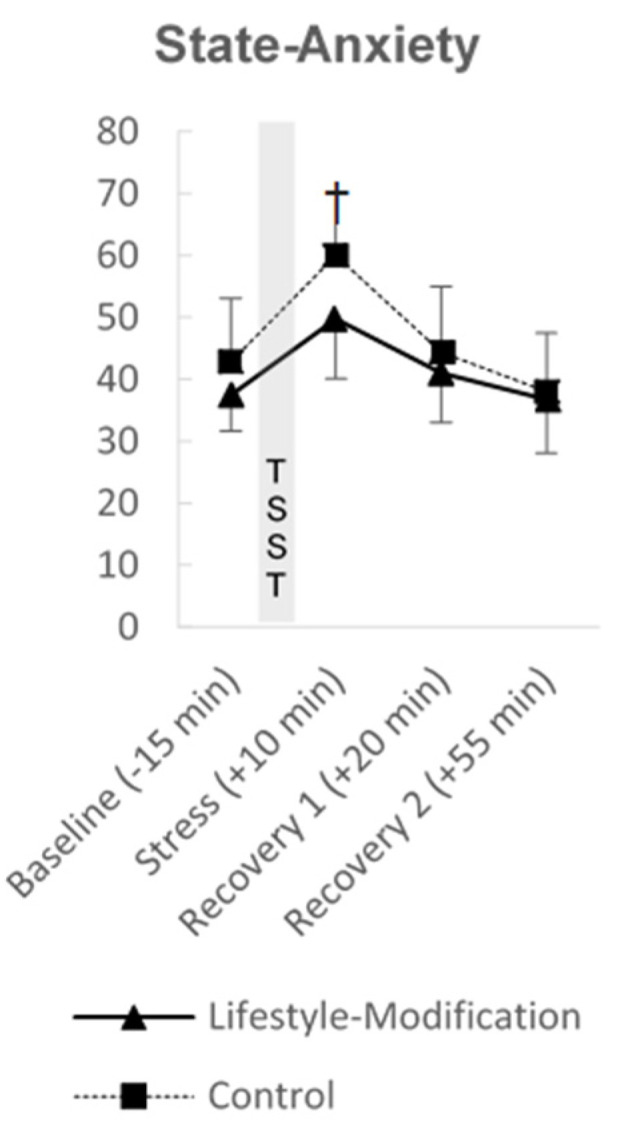
Results of the exploratory sub-study: State anxiety as measured with the state version of the State-Trait Anxiety Inventory in patients of the lifestyle-modification group (*n* = 8) and controls (*n* = 8). Results are expressed as mean ± standard deviation. † *p* < 0.10.

**Table 1 jcm-10-05070-t001:** Sociodemographic and clinical characteristics (mean ± standard deviation (range)), if not indicated otherwise.

	Lifestyle Modification (*n* = 47)	Lifestyle Modification TSST (*n* = 8)	Control (*n* = 50)	Control TSST (*n* = 8)
Age years	50.28 ± 11.90 (18–74)	44.6 ± 14.3 (18–57)	45.54 ± 12.49 (19–71)	49.25 ± 4.30 (43–57)
Female *n* (%)	34 (72.3)	8 (100)	35 (70)	8 (100)
Weight	72.79 ± 14.90 (52–100)	71.13 ± 12.87 (62–97)	70.24 ± 16.86 (49.6–150)	61.19 ± 9.72 (51–79)
Height	171.19 ± 9.05 (152–196)	173.88 ± 6.11 (167–183)	173.76 ± 9.94 (156–197)	169.00 ± 4.00 (164–176)
Anamnestic pattern *n* (%)				
Proctitis	14 (29.8)	1 (12.5)	15 (30)	3 (37.5)
Left-sided colitis	17 (36.2)	4 (50)	15 (30)	2 (25)
Pancolitis	13 (27.7)	3 (37.5)	17 (34)	3 (37.5)
Missing	3 (6.4)	0 (0)	3 (6)	0 (0)
Time since diagnosis years	18.04 ± 12.00 (2–46)	10.50 ± 7.95 (2–24)	14.76 ± 10.99 (1–43)	12.63 ± 4.03 (5–17)
Prior integrative medicine inpatient treatment at Kliniken Essen-Mitte *n* (%)	13 (27.7)	2 (25)	12 (24)	1 (12.5)
Prior integrative medicine day-care treatment at Kliniken Essen-Mitte *n* (%)	7 (14.9)	2 (25)	3 (6)	1 (12.5)
Smokers *n* (%)	2 (4.3)	1 (12.5)	3 (6)	1 (12.5)
Married *n* (%)	33 (70.2)	6 (75)	39 (78)	6 (75)
Education *n* (%)				
Secondary school	17 (36.1)	3 (37.5)	11 (22)	3 (37.5)
High school (“Abitur”)	12 (25.6)	4 (50)	14 (28)	2 (25)
University degree	18 (38.3)	1 (12.5)	25 (50)	3 (37.5)
Medication intake *n* (%)				
Steroids, Azathioprine, Biologicals	9 (19.2)	4 (50)	7 (14)	1 (12.5)
Mesalazine	33 (70.2)	7 (87.5)	34 (68)	7 (87.5)
Herbal medicine	7 (14.9)	0 (0)	15 (30)	1 (12.5)
Other	8 (17)	0 (0)	12 (24)	2 (25)

Note. TSST = Trier Social Stress Test.

**Table 2 jcm-10-05070-t002:** Long-term effects of the comprehensive lifestyle-modification program on stress, anxiety, depression and flourishing (mean ± standard deviation (range)).

	*n*	Baseline	Week 12	Week 24	Week 48	Group Differences Week 12		Group Differences Week 24		Group Differences Week 48	
						*p*	η^2^_p_	*p*	η^2^_p_	*p*	η^2^_p_
**Perceived Stress (PSS-10)**											
Lifestyle-modification	47	22.30 ± 5.60	14.00 ± 6.38	15.76 ± 6.44	13.75 ± 7.20	<0.001	0.148	0.023	0.054	0.060	0.037
Control	50	22.20 ± 6.66	18.59 ± 6.89	18.47 ± 6.29	16.05 ± 6.80						
**Anxiety (HADS_A)**											
Lifestyle-modification	47	10.30 ± 4.33	6.67 ± 3.84	7.61 ± 4.26	6.46 ± 3.98	0.075	0.033	0.824	0.001	0.627	0.003
Control	50	9.81 ± 4.16	7.45 ± 3.55	7.55 ± 3.48	6.55 ± 3.20						
**Depression (HADS_D)**											
Lifestyle-modification	47	6.78 ± 3.84	4.74 ± 3.39	5.57 ± 3.52	4.45 ± 3.46	0.146	0.022	0.430	0.007	0.764	0.001
Control	50	7.00 ± 3.48	5.81 ± 3.91	6.18 ± 3.59	4.74 ± 3.36						
**Flourishing**											
Lifestyle-modification	36	42.06 ± 8.69	43.77 ± 6.32	-	45.59 ± 6.11	0.247	0.018	-	-	0.006	0.100
Control	41	41.98 ± 5.89	42.41 ± 5.24	-	41.65 ± 6.26						

Note. *p*-values are based on univariate analyses of covariance with ‘group’ as the between-subject factor and baseline values as covariates. HADS_A/HADS_D = Hospital Anxiety and Depression Scale; PSS = Perceived Stress Scale.

## Data Availability

Data is not available due to ethical reasons and privacy issues.

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
