# Peer review of "Comprehensive Lifestyle Modification Influences Medium-Term and Artificially Induced Stress in Ulcerative Colitis—A Sub-Study within a Randomized Controlled Trial Using the Trier Social Stress Test"

_jcm, 2021, doi:10.3390/jcm10215070_

Round 1
Reviewer 1 Report
Koch et al. analyzed the effect of a comprehensive lifestyle modification program on stress in patients with quiescent ulcerative colitis. The group already published the results of this study (with some identical figures/tables about the patient cohort) in the same journal in regards of health-related quality of life, disease activity, GI symptoms and microbiome.
Strengths of this study are the randomized controlled design and the use of a standardized protocol to evaluate stress response including stress hormones, pulse, white blood cell count and blood pressure (Tier Social Stress Test, TSST).
However, there are considerable flaws in the statistical analysis of the TSST. The p-value has been set at 0.1 and no correction for multiple comparison has been applied. The authors compare many variables at different time points with a small sample size (n=8 in each group) and absolutely no correction for multiple comparison, which leads to an inflation in Type I error. According to the E 9 ICH Statistical Principles for Clinical Trials guidelines this is not appropriate. Judging from the presented graphs (Fig. 2,3,4) the reported conclusions are clearly not supported by the data (i.e., difference in ACTH levels at timepoint 'Recovery 2', line 228, Fig 2). In line 188 authors mention missing values, how many values were missing and in which context?
Additionally, there were considerable differences between the Lifestyle-modification (LSM) TSST and Control TSST group in terms of disease pattern and medication (Table. 1). Fifty percent of the LSM TSST group had Steroids, Azathioprine or Biologicals compared to 12.5 % of the control group. These medications can affect the concentration of hormones and immune cells and thus TSST read out. Indeed, a baseline decreased lymphocyte concentration and increased neutrophils are evident in the LSM TSST group (Fig. 2), an effect that could be explained by corticosteroids which needs to be discussed appropriately.
Lifestyle modification and treatment of 'functional' symptoms in ulcerative colitis patients in remission is an important topic. However, given the above-mentioned flaws in statistical analysis the main conclusions of the paper cannot be supported. One third of the citations in this work are self-citations, a high number in the field of IBD research. I would advise the authors of this manuscript to reach out to a statistician and use the (promising) TSST data of this manuscript as a starting point for a proper sample size estimation and expand sample size accordingly.
Author Response
Point 1: Koch et al. analyzed the effect of a comprehensive lifestyle modification program on stress in patients with quiescent ulcerative colitis. The group already published the results of this study (with some identical figures/tables about the patient cohort) in the same journal in regards of health-related quality of life, disease activity, GI symptoms and microbiome.
Strengths of this study are the randomized controlled design and the use of a standardized protocol to evaluate stress response including stress hormones, pulse, white blood cell count and blood pressure (Tier Social Stress Test, TSST).
However, there are considerable flaws in the statistical analysis of the TSST. The p-value has been set at 0.1 and no correction for multiple comparison has been applied. The authors compare many variables at different time points with a small sample size (n=8 in each group) and absolutely no correction for multiple comparison, which leads to an inflation in Type I error. According to the E 9 ICH Statistical Principles for Clinical Trials guidelines this is not appropriate.
Response 1: To compare the effects on TSST, a sub-sample (n = 16), was analysed with the p-value set at p < .10 but with at a lower confidence level. Due to the explorative nature of the sub-study and small sample size, no adjusted p value for multiple testing was applied. This small sample size is clearly a study limitation and hence no firm conclusions can be made. We are aware of this limitating issue which is why currently we are conducting a study at the clinic also using the TSST to replicate the present results. This new study is also powered for the TSST. We will be happy to make it even clearer in a revised manuscript version that the TSST data are purely exploratory preliminary work for the larger TSST study.
Point 2: Judging from the presented graphs (Fig. 2,3,4) the reported conclusions are clearly not supported by the data (i.e., difference in ACTH levels at timepoint 'Recovery 2', line 228, Fig 2).
Response 2: Since we used ANCOVA as the method of analysis, group differences in the corresponding variables could be controlled for baseline, and this influence was thus "calculated out". This results in significant differences that may not appear so in a graphical representation,
Point 3: In line 188 authors mention missing values, how many values were missing and in which context?
Response 3: There were the following missuing values which have been replaced by means of the Markov chain Monte Carlo procedure: At baseline, there were missing 1x ACTH, 1x leukocytes, 1x thrombocytes, 1x monocytes, 1x lymphocytes, 1x STAI item 3. At timepoint stress, there were missing: 1x ACTH, 1x pulse, 1x leukocytes, 1x thrombocytes, 1x monocytes, 1x lymphocytes, 1x neutrophils (all from other patients than at timepoint baseline). At timepoint recovery 1, there were missing: 1x ACTH2, 1x diastolic blood pressure, 1x systolic blood pressure, 1x pulse, 2x leukocytes, 2x thrombocytes, 2x lymphocytes, 2x monocytes, 2x neutrophils. At timepoint post 3, 1x ACTH, 2x pulse,1x leukocytes, 1x thrombocytes, 1x lymphocytes, 1x monocytes, 1x neutrophils.
Point 4: Additionally, there were considerable differences between the Lifestyle-modification (LSM) TSST and Control TSST group in terms of disease pattern and medication (Table. 1). Fifty percent of the LSM TSST group had Steroids, Azathioprine or Biologicals compared to 12.5 % of the control group. These medications can affect the concentration of hormones and immune cells and thus TSST read out. Indeed, a baseline decreased lymphocyte concentration and increased neutrophils are evident in the LSM TSST group (Fig. 2), an effect that could be explained by corticosteroids which needs to be discussed appropriately.
Response 4: Due to the small subsample for the TSST, there were indeed group differences from baseline. Since we used ANCOVA as the method of analysis, group differences in the corresponding variables could be controlled for baseline, and this influence was thus "calculated out". In the current larger TSST study, these problems will be addressed.
Point 5: Lifestyle modification and treatment of 'functional' symptoms in ulcerative colitis patients in remission is an important topic. However, given the above-mentioned flaws in statistical analysis the main conclusions of the paper cannot be supported. One third of the citations in this work are self-citations, a high number in the field of IBD research. I would advise the authors of this manuscript to reach out to a statistician and use the (promising) TSST data of this manuscript as a starting point for a proper sample size estimation and expand sample size accordingly.
Response 5: Thank you for the constructive suggestions. We are aware that the sample is much too small to be considered confirmatory. That is why it is clearly stated in several places that these are purely exploratory analyses. However, a larger sample was unfortunately not possible within the framework of the underlying RCT. Currently, we are conducting a study at the clinic also using the TSST to replicate the present results. This new study is also powered for the TSST. In addition, we are including healthy controls in this new study to add further value. We will be happy to make it even clearer in a revised manuscript version that the TSST data are purely exploratory preliminary work for a larger TSST study. We are happy to revise the cited papers again.
Reviewer 2 Report
The study by Koch and coworkers presents results regarding patients with UC undergoing a stressful stimulus. The study has some relevant method concerns. The introduction is clear enough but, in my opinion, the authors refer to recent literature without citing the related papers. Line 79: "There are studies on acute stress". What studies are considered by the authors? The reader cannot understand which previous results are considered relevant by the authors. Again: prefrontal-amygdala complex is cited but references to fMRI works that have studied the amygdala in UC patients are missing. Considering the focus of the paper, the literature on stress habituation in IBD patients is particularly relevant. Methods: Authors must identify who selected patients. Were they consecutive patients? Selected? Who and how was it established that the patients were in remission? Was only the CAI clinical index used? Who determined if psychological problems were present? Based on a psychiatric or psychological visit? Who made this hypothetical visit? If patients with depression were excluded, why administer a depression questionnaire? The authors must explain these points. A particularly important point is the current or past presence of psychotherapies in enrolled patients. Did the authors ask the participants if they had had psychotherapy or were they undergoing psychotherapy? Discussion The discussion suffers from the lack of comparison with the literature neglected by the authors. The authors acknowledge the low numbers of patients enrolled and that their work is preliminary, yet a conflict with the literature that addresses the topic of psychological interventions in IBD.
Author Response
Point 1: The study by Koch and coworkers presents results regarding patients with UC undergoing a stressful stimulus. The study has some relevant method concerns. The introduction is clear enough but, in my opinion, the authors refer to recent literature without citing the related papers. Line 79: "There are studies on acute stress". What studies are considered by the authors? The reader cannot understand which previous results are considered relevant by the authors. Again: prefrontal-amygdala complex is cited but references to fMRI works that have studied the amygdala in UC patients are missing. Considering the focus of the paper, the literature on stress habituation in IBD patients is particularly relevant.
Response 1: Thank you very much for your constructive comment. We would be happy to add the aspects and literature references you mentioned to our introduction and into our discussion.
Point 2: Methods: Authors must identify who selected patients. Were they consecutive patients? Selected?
Response 2: Patients were recruited through newspaper and online announcements via a study announcement advertised by the German Crohn’s Colitis Organization, as well as the Department of Internal and Integrative Medicine at the Kliniken Essen-Mitte, Essen, Germany. Patients were randomised using stratified block randomization (Strata: sex, azathioprine and biologics) to either the comprehensive lifestyle-modification program group or the control group. Regarding TSST, all patients were informed about the experiment and could choose whether they wanted to participate. Female patients (due to the different hormonal responses to stress between the sexes only females) were then included on a voluntary basis.
Point 3: Who and how was it established that the patients were in remission? Was only the CAI clinical index used?
Response 3: At the beginning of the study, all patients underwent a medical visit on site. In this context, the disease activity was also determined by means of Clinical Activity Index (Rachmilewitz. 1989). Patients diagnosed with ulcerative colitis who had been in clinical remission for no longer than12 months at the longest according to the CAI were eligible.
Point 4: Who determined if psychological problems were present? Based on a psychiatric or psychological visit? Who made this hypothetical visit?
Response 4: During the screening visit, a medical interview was conducted with each individual patient. In this context, it was also asked whether the patients suffer from a mental illness or are undergoing psychotherapeutic treatment.
Point 5: If patients with depression were excluded, why administer a depression questionnaire? The authors must explain these points.
Response 5: Only patients suffering from severe depression requiring treatment were excluded from the study. However, the HADS is also very well suited to measure depression and anxiety at a subclinical level. Moreover, it is a very common questionnaire in the clinical context, which is why it was used here.
Point 6: A particularly important point is the current or past presence of psychotherapies in enrolled patients. Did the authors ask the participants if they had had psychotherapy or were they undergoing psychotherapy?
Response 6: Severe psychological illness requiring treatment like depression was an exclusion criteria. However, it is not possible to say whether the patients had received psychotherapeutic treatment earlier in their lives.
Point 7: Discussion The discussion suffers from the lack of comparison with the literature neglected by the authors. The authors acknowledge the low numbers of patients enrolled and that their work is preliminary, yet a conflict with the literature that addresses the topic of psychological interventions in IBD.
Response 7: In the revised version of our manuscript we would like to expand the discussion in this regard and also include the newly added literature from the introduction.
Round 2
Reviewer 2 Report
No further comment